# Ferroptosis Inhibitory Compounds from the Deep-Sea-Derived Fungus *Penicillium* sp. MCCC 3A00126

**DOI:** 10.3390/md21040234

**Published:** 2023-04-10

**Authors:** You-Jia Hao, Zheng-Biao Zou, Ming-Min Xie, Yong Zhang, Lin Xu, Hao-Yu Yu, Hua-Bin Ma, Xian-Wen Yang

**Affiliations:** 1College of Marine Sciences, Shanghai Ocean University, 999 Hucheng Ring Road, Shanghai 201306, China; 2Key Laboratory of Marine Genetic Resources, Third Institute of Oceanography, Ministry of Natural Resources, 184 Daxue Road, Xiamen 361005, China; 3Institute of Drug Discovery Technology, Ningbo University, Ningbo 315211, China

**Keywords:** deep sea, fungus, *Penicillium* sp., cytotoxicity, necroptosis

## Abstract

Two new xanthones (**1** and **2**) were isolated from the deep-sea-derived fungus *Penicillium* sp. MCCC 3A00126 along with 34 known compounds (**3**–**36**). The structures of the new compounds were established by spectroscopic data. The absolute configuration of **1** was validated by comparison of experimental and calculated ECD spectra. All isolated compounds were evaluated for cytotoxicity and ferroptosis inhibitory activities. Compounds **14** and **15** exerted potent cytotoxicity against CCRF-CEM cells, with IC_50_ values of 5.5 and 3.5 μM, respectively, whereas **26**, **28**, **33**, and **34** significantly inhibited RSL3-induced ferroptosis, with EC_50_ values of 11.6, 7.2, 11.8, and 2.2 μM, respectively.

## 1. Introduction

The oceans cover over 70% of the world’s surface, with 95% of them being deeper than 1000 m. In recent years, nearly half of the new marine natural products (MNPs) have been isolated from marine microorganisms [1,2,3], especially fungi, the most diverse and abundant eukaryotes on Earth, which can be distributed in any currently known extreme environment [4]. As a region rarely explored, the deep sea is characterized by a high pressure, a low/high (such as hydrothermal mouth) temperature, a high salt concentration, the absence of light, oligotrophic conditions, a high halogen content, and so on. To adapt to such extreme environments, deep-sea-derived microorganisms must develop special metabolic mechanisms, giving rise to tremendous secondary metabolites with unique structures and potent bioactivities [5]. For more than half a century, MNPs have been continuously discovered, but those from the deep sea are rare [6,7]. In recent years, with the development of deep-sea sample collection technology, reports of deep-sea MNPs have increased significantly. As an important group of deep-sea microorganisms, fungi can produce a large number of structurally novel and biologically active secondary metabolites, which have attracted extensive attention from researchers. For example, vercytochalasins A and B are two novel, biosynthetically related cytochalasins isolated from *Curvularia verruculosa*, the endophytic fungus of the deep-sea lobster *Shinkaia crosnieri*. Vercytochalasin A is the most potent natural product against angiotensin-I-converting enzyme (ACE), with an IC_50_ value of 505 nM [8]. Chevalinulins A and B are two indole alkaloids with a rare spiro-[bicyclo[2.2.2]octane-diketopiperazine] skeleton. They both exhibit significant in vivo proangiogenic activity in transgenic zebrafish [9]. 

Xanthones, also known as 9*H*-xanthen-9-ones, are a class of yellow compounds bearing a dibenzo-γ-pyrone scaffold. They are widely distributed in plants, lichens, and microorganisms of terrestrial and marine origin, and exhibit diverse biological activities such as antiviral [10], cytotoxic [11], antibacterial [12], antifungal [13], and hypoglycemic [14] activities. The molecular skeleton of xanthones can bind with a variety of targets, so this family of compounds is often called “privileged structures” [15]. They are regarded as typical aromatic polyketone and appear in the form of fully aromatic or hydrogenated derivatives [16]. In general, xanthones can be classified into monomers, dimers, and heterodimers. According to the degree of hydrogenation of the skeleton aromatic ring, they can be further split into four subclasses: fully aromatic xanthones, dihydro-, tetrahydro-, and hexahydro-xanthones [16]. From 2010 to 2021, 100 marine xanthones were reported, among which 51 were new compounds. Most of the new xanthones were derived from marine fungi, especially deep-sea fungi isolated from deep-sea sediments or organisms [17]. Therefore, deep-sea-derived fungi are undoubtedly an important resource for the discovery of xanthones with novel structures and significant bioactivities.

During our ongoing search for bioactive secondary metabolites from deep-sea-derived fungi [18,19,20,21], the crude extract of *Penicillium* sp. MCCC 3A00126 isolated from the Eastern Pacific Ocean at a depth of 5246 m showed significant cytotoxicity against acute lymphoblastic leukemia CCRF-CEM with the cell survival rate of 29.8 % under the concentration of 10 μg/mL. Therefore, it was subjected to a systematic chemical investigation. As a result, two novel (**1** and **2**) and 13 known (**3**–**15**) xanthones were isolated and purified, along with 21 known miscellaneous compounds (**16**–**36**) (Figure 1). Herein, the details of isolation, structure, and bioactivity are reported.

## 2. Results and Discussion

Compound **1** was obtained as a colorless gum. The molecular formula C_18_H_18_O_9_ was determined by the positive HR-ESI-MS (high resolution electrospray mass spectrometry spectrum) at *m*/*z* 401.0839 [M + Na]^+^, suggesting ten degrees of unsaturation. The ^1^H (Appendix A in the Appendix A) and ^13^C (Appendix A in the Appendix A) NMR (nuclear magnetic resonance) spectroscopic data (Table 1) exhibited the presence of one methoxyl [*δ*_H_ 3.84 (3H, s); *δ*_C_ 53.4 q], one methyl singlet [*δ*_H_ 2.16 (3H, s); *δ*_C_ 20.8 q], one oxygenated [*δ*_H_ 5.13 (2H, s); *δ*_C_ 65.0 t], and two aliphatic [*δ*_H_ 2.25 (2H, m), 2.87 (2H, m); *δ*_C_ 24.2 t, 26.1 t] methylene groups, one oxygenated aliphatic [*δ*_H_ 4.09 (1H, dd, *J* = 10.3, 3.6 Hz); *δ*_C_ 72.6 d] and two olefinic [*δ*_H_ 6.75 (1H, s), 6.86 (1H, s); *δ*_C_ 109.9 d, 105.5 d] methines, as well as ten quaternary carbons including one oxygenated aliphatic (*δ*_C_ 76.2 s), six olefinic (*δ*_C_ 109.6 s, 117.0 s, 145.0 s, 156.1 s, 160.7 s, 167.4 s), and three carbonyl (*δ*_C_ 170.5 s, 172.7 s, 182.0 s] carbons. These signals were very similar to aspergillusone B (**4**) [22], except for an additional acetyl group [*δ*_H_ 2.16 (3H, s); *δ*_C_ 20.8 q, 170.5 s] at the C-11 position. This was supported by the downfield shifts of H-11 from δ_H_ 4.76 to δ_H_ 5.13 and C-11 from δ_C_ 64.4 to δ_C_ 65.0. Further confirmation was obtained by the HMBC (heteronuclear multiple-bond correlation) correlations of H_2_-11 to the carbonyl group of the acetyl moiety; and the ^1^H-^1^H COSY (correlation spectroscopy) cross peaks of H_2_-6 to H_2_-5/H-7 (Figure 2).

The coupling constants between H-7 and H_2_-6 of **1** (*J* = 10.3 Hz, 3.6 Hz) indicated the same pseudoaxial position of H-7 as that of **4** [22], as it was found in a simple MM2 conformational study of both possible 7,8-anti and 7,8-syn diols (Figure 2). The observed optical rotation value of **1** ([*α*${]}_{D}^{25}$ −82.5) was close to that of **4** ([*α*${]}_{D}^{25}$ −46.3) in the same concentration (*c* 0.2) and the same solvent (CHCl_3_), (*c* 0.2, CHCl_3_), suggesting they have the same absolute configuration at C-7 and C-8. For the further confirmation, the ECD (electron circular dichroism) spectra were calculated for (7*R*,8*R*)-**1** (**1a**) and its enantiomer (7*S*,8*S*)-**1** (**1b**) using Yinfo Cloud Computing Platform (https://cloud.yinfotek.com, accessed on 13 June 2022). Thirty states of each seven conformers were calculated to generate the ECD curves. As shown in Figure 3, the calculated ECD spectrum of **1a** was consistent with that of the experimental one. On the basis of the above evidence, compound **1** was then elucidated as 11-*O*-acetylaspergillusone B.

Compound **2** was obtained as a amorphous yellow solid. The molecular formula C_17_H_12_O_7_ was established by its positive HR-ESI-MS spectrum at *m*/*z* 351.0482 [M + Na]^+^. The ^1^H and ^13^C data of **2** showed the presence of two methoxyls, five methines and ten quaternary carbons, which were closely related to those of huperxanthone A (**13**) [14], except that the *sp*^2^ quaternary carbon at C-7 (*δ*_C_ 151.1 s) in **13** was changed as an *sp*^2^ methine (*δ*_C_ 123.1 d) in **2**. By detailed analysis of its HSQC (heteronuclear single quantum correlation), ^1^H–^1^H COSY, HMBC, and NOESY (nuclear Overhauser effect) spectroscopic data, compound **2** was then established as 7-dehydroxyhuperxanthone A.

By comparison of the NMR and MS data with those published in the literature, 34 known compounds were identified as 13 xanthones: (7*R*,8*R*)-*α*-diversonolic ester (**3**) [23,24], aspergillusone B (**4**) [22], 8-hydroxy-6-methyl-9-oxo-9H-xanthene-1-carboxylate (**5**) [25], yicathin B (**6**) [26], pinselin (**7**) [27], sydowinins A (**8**) and B (**9**) [28], huperxanthone C (**10**) [14], 13-*O*-acetylsydowinin B (**11**) [29], 8-(methoxycarbonyl)-1-hydroxy-9-oxo-9H-xanthene-3-carboxylic acid (**12**) [25], huperxanthone A (**13**) [14], sterigmatocystin (**14**) [30], 5-methoxysterigmatocystin (**15**) [31]; six anthraquinones: versicolorin B (**16**) [32], 8-*O*-methylversicolorin B (**17**) [33], anthraquinone aversin (**18**) [34], averufin (**19**) [35], 6-*O*-methylaverufin (**20**) [36], questin (**21**) [37]; five sesquiteroenoids: (*S*)-(+)-sydonic acid (**22**) [38], (*S*)-(+)-11-dehydrosydonic acid (**23**) [38], (−)-12-acetoxy-1-deoxysydonic acid (**24**) [39], (7*S*,11*S*)-(+)-12-acetoxysydonic acid (**25**) [38], (−)-(7*R*,10*R*)-*iso*-10-hydroxysydowic acid (**26**) [39]; four diphenyl ethers: diorcinol (**27**) [40], verticilatin (**28**) [41], (*R*)-3-((2-(2-hydroxypropan-2-yl)-6-methyl-2,3-dihydrobenzofuran-4-yl)oxy)-5-methylphenol (**29**) [42], (3*S*)-3,4-dihydro-5-(3-hydroxy-5-methylphenoxy)-2,2,7-trimethyl-2*H*-chromen-3-ol (**30**) [43]; one polyketone: 3-hydroxy-5-methylphenyl-2,4-dihydroxy-6-methylbenzoate (**31**) [44]; four indole alkaloids: brevianamide F (**32**) [45], notoamide I (**33**) [46], notoamide C (**34**) [47], psychrophilin D (**35**) [48], and one steroid: 5*a*,8*a*-epidioxy-22*E*-ergosta-6,9(11)-trien-3*β*-ol (**36**) [49]. 

Since the crude extract of *Penicillium* sp. MCCC 3A00126 showed a potent anti-proliferative effect on CCRF-CEM, all 36 isolates were subjected to cytotoxicity tests on the same acute lymphoblastic leukemia using the CCK-8 assay. As shown in Figure 4, under a concentration of 20 μM, two compounds, **14** and **15**, exerted potent activity, with cell survival rates of 6.2% and 7.3%, respectively, while seven compounds, **4**, **8**, **11**, **17**, **20**, **28**, and **29**, showed weak effects, with cell survival rates of 70.2%, 78.5%, 78.8%, 62.3%, 75.8%, 55.3%, and 55.3%, respectively. Interestingly, compounds **14** and **15** possess a difuran ring at C-5 and C-6, which might be the key to the bioactivity.

Compounds **14** and **15** were further evaluated to determine their 50% inhibiting concentration (IC_50_) against CCRF-CEM using five different concentrations: 1 μM, 2.5 μM, 5.0 μM, 10.0 μM, and 20.0 μM. The IC_50_ values of **14** and **15** were found to be 5.5 μM and 3.5 μM, respectively (Figure 5). 

Ferroptosis is an iron-dependent mode of necroptosis induced by certain small molecules, such as RSL3 (the glutathione peroxidase 4 inhibitor), which is different from apoptosis, necrolysis, and autophagy [50]. Its main characteristics are the generation of ROS (reactive oxygen species), LPO (lipid peroxidation), and iron accumulation. RSL3 acts on specific targets in cells and causes a reduction in antioxidants GSH (glutathione) and GPX4 (glutathione peroxidase 4), resulting in the accumulation of ROS in cells, LPO in cells, and ferroptosis in cells under the synergistic effect of iron [51]. Many tumor cells that are easy to metastasize are prone to ferroptosis, so inducing and inhibiting ferroptosis for pharmacological regulation has great potential in the treatment of certain cancers.

To further investigate whether these isolates could inhibit ferroptosis, RSL3, the GPX4 inhibitor, was used to induce ferroptosis in CCRF-CEM cells. As a result, compounds **26**, **28**, **33**, and **34** exerted strong inhibition, with cell survival rates of 83.9%, 110.0%, 99.0%, and 105.2%, respectively, under a concentration of 20 μM. Additionally, compounds **3**, **27**, **29**, and **30** showed weak activity, with cell survival rates of 36.0%, 16.6%, 19.5%, and 28.8%, respectively (Figure 6). 

To determine the 50% effective concentration (EC_50_) of compounds **26**, **28**, **33**, and **34**, four different concentrations (1 μM, 5.0 μM, 10.0 μM, and 20.0 μM) were adopted on RSL3-induced ferroptosis in CCRF-CEM cells, providing corresponding IC_50_ values of 11.6 μM, 7.2 μM, 11.8 μM, and 2.2 μM, respectively (Figure 7). 

As ferroptosis was triggered by lipid peroxidation, many ferroptosis inhibitors exhibited antioxidant activity, such as ferrostain-1 (Fer-1) [50,52,53]. Therefore, the 2,2-diphenyl-1-picrylhydrazyl (DPPH) assay was performed on these compounds to evaluate their intrinsic antioxidant potential. However, none showed positive effects under a concentration of 20 μM (Figure 8), indicating that compounds **26**, **28**, **33**, and **34** might exert ferroptosis inhibition by other mechanisms instead of DPPH. 

## 3. Materials and Methods

### 3.1. General Experimental Procedures

The HR-ESI-MS spectra were obtained on a Waters Xevo G2 Q-TOF mass spectrometer equipped with a Spray™ ESI source in both the positive and negative ion mode. NMR spectra were recorded in CDCl_3_, CD_3_OD, or DMSO-*d*_6_ on a Bruker Avance III 400 Mz spectrometer at room temperature. Optical rotation was measured by an Anton Paar MCP 100 polarimeter. UV and ECD spectra were acquired on a JASCO J-810 spectropolarimeter. Preparative HPLC (high-performance liquid chromatography) separations for purification were carried out on an Agilent 1260 system with a semi-preparative chromatographic column (COSMOSIL 5C_18_-MS-II, 5PFP, SL-II, 250 mm × 10 mm). Materials for column chromatography (CC) included silica gel, ODS (octadecylsilyl), and Sephadex LH-20.

### 3.2. Biological Material

The deep-sea-derived fungus *Penicillium* sp. MCCC 3A00126 was isolated from a sediment sample collected from the Eastern Pacific Ocean at a depth of 5246 m by Professor De-Zan Ye of the Third Institute of Oceanography, Ministry of Natural Resources, in September 2003. The 18S rRNA gene sequence alignment (AM18217) showed great similarity (99.88%) to *Penicillium* sp. PB g (GenBank accession number MK372218.1); therefore, it was identified to be *Penicillium* sp. It was deposited at the Marine Culture Collection of China (Xiamen, China) with accession number MCCC 3A00126.

### 3.3. Fermentation, Isolation, and Purification

The strains stored at −80 °C were inoculated into a 250 mL conical flask containing 100 mL PDB culture medium to conduct initial activation for three days in a shaking table culture at 28 °C. Then, under the same culture conditions, 1 mL of the fungal liquid was placed in another 1 L conical flask containing 250 mL of PDB medium to conduct secondary activation.

The secondary activated fungal strain was inoculated into 60 Erlenmeyer flasks containing 80 g rice and 120 mL distilled water. The fermentation was kept under static conditions at 25 °C. After 40 days, 400 mL of EtOAc was added to each flask over 12 h. The organic solvent was filtered. The procedure was repeated four times. The organic solvents were combined and concentrated to a small volume. The latter was dissolved in MeOH and extracted by PE (petroleum ether) three times. The MeOH layer was concentrated to provide a crude extract (26.3 g), which was subjected to MPLC (medium-pressure liquid chromatography) over silica gel using the CH_2_Cl_2_/MeOH gradient (100% → 70%) to obtain four fractions (Fr.1–Fr. 4). Fraction Fr.1 (0.46 g) was separated by repeated column chromatography (CC) over ODS (MeOH/H_2_O) and Sephadex LH-20 (CH_2_Cl_2_/MeOH, 1:1 and 0:1), then purified by semi-prep. HPLC (C18, 10 × 250 mm, MeOH/H_2_O, 80% → 100%), providing **2** (15.0 mg) and **5** (14.0 mg). Fraction Fr.2 (1.1 g) was subjected to CC over ODS and Sephadex LH-20, then purified by semi-prep. HPLC (C18, 10 × 250 mm, MeOH/H_2_O, 70% → 90%), yielding **14** (2.0 mg), **15** (3.0 mg), and **18** (5.4 mg). Fraction Fr.3 (3.7 g) was separated by ODS CC with MeOH/H_2_O (30% → 80%, 3 h, 80% → 90%, 3 h) to obtain thirteen subfractions (Fr.3.1–Fr.3.13). Subfractions (Fr.3.1–Fr.3.9) were subjected to CC over Sephadex LX-20 using MeOH to yield compounds **3** (2.0 mg), **10** (9.0 mg), **11** (4.0 mg), **13** (8.0 mg), **16** (2.0 mg), **19** (2.5 mg), **20** (6.0 mg), **28** (9.5 mg), and **36** (28.0 mg), respectively. Sub-fraction Fr.3.10 was subjected to CC over Sephadex LH-20 (CH_2_Cl_2_/MeOH, 1:1) and silica gel (CH_2_Cl_2_/MeOH, 45:1), followed by separation using HPLC (C18, 10 mm × 250 mm, MeOH/H_2_O, 80% → 90%) to give **8** (15.0 mg) and **9** (2.0 mg). Fr.3.11 was separated by CC over LH-20 (MeOH) and silica gel (CH_2_Cl_2_/MeOH, 55:1) to obtain **12** (1.0 mg). Fr.3.12 was purified by semi-prep. HPLC (PFP-pentafluorophenyl group, 10 mm × 250 mm, MeOH/H_2_O, 60% → 90%), yielding **21** (3.0 mg), **29** (6.6 mg), and **30** (1.3 mg). Fr.3.13 was further separated by HPLC (C18, 10 mm × 250 mm, MeOH/H_2_O, 55% → 85%) to yield **6** (3.0 mg). 

Using similar procedures, fraction Fr.4 (3.3 g) was separated into thirteen subfractions (Fr.4.1–Fr. 4.13) by CC over ODS (MeOH/H_2_O, 5 → 60%, 4 h, 60% → 100%, 2 h). Compounds **1** (8.0 mg), **4** (23.3 mg), **17** (23.0 mg), **22** (39.0 mg), **27** (7.0 mg), and **31** (19.8 mg) were obtained from Fr.4.1–Fr.4.6 by CC over Sephadex LH-20 (MeOH). Fr.4.7 and Fr.4.8 were separated by HPLC (C18, 10 mm × 250 mm, MeOH/H_2_O, 35% → 65%) to obtain **26** (1.0 mg) and **32** (2.6 mg), respectively. Fr.4.9 was subjected to Sephadex LH-20 CC, eluting with MeOH, and then purification by HPLC (C18, 10 mm × 250 mm, MeOH/H_2_O, 55% → 85%) afforded **24** (3.0 mg) and **25** (9.5 mg). Fr.4.10 was separated by HPLC on silica gel (10 mm × 250 mm, CH_2_Cl_2_/MeOH, 100% → 90%) to yield **35** (12.0 mg). Compounds **23** (2.5 mg) and **33** (1.7 mg) were also separated from Fr.4.11 using HPLC (C18, 10 mm × 250 mm, MeOH/H_2_O, 60% → 80%). Fr.4.12 was chromatographed by CC over silica gel (PE/EtOAc, 3:1) to yield **7** (3.2 mg) and Fr.4.13 was subjected HPLC (C18, 10 mm × 250 mm, MeOH/H_2_O, 55% → 85%) to yield **34** (0.7 mg).

11-*O*-Acetylaspergillusone B (**1**). Colorless gum; [*α*${]}_{D}^{25}$ −82.5 (*c* 0.2, CHCl_3_), −19.0 (*c* 0.1, MeOH); UV (2.6 mM, MeOH) *λ*_max_ (log *ε*) 212 (2.32), 240 (2.42), 289 (1.69), 331 (2.04) nm; ECD (2.6 mM, MeOH) *λ*_max_ (Δ*ε*) 216 (1.99), 263 (2.06), 313 (1.79); ^1^H and ^13^C NMR data, Table 1; HRESIMS *m*/*z* 401.0839 [M + Na]^+^ (calcd for C_18_H_18_O_9_Na, 401.0849).

7-Dehydroxyhuperxanthone A (**2**). Yellow amorphous solid; UV (3.0 mM, MeOH) *λ*_max_ (log *ε*) 266 (3.23); ^1^H and ^13^C NMR data, Table 1; HRESIMS *m*/*z* 351.0482 [M + Na]^+^ (calcd for C_17_H_12_O_7_Na, 351.0481).

### 3.4. ECD Calculation

ECD calculations were performed using Yinfo Cloud Computing Platform, a user-friendly and versatile web server for biomedicinal, material, and statistical research (https://cloud.yinfotek.com, accessed on 13 June 2022). The conformational analysis of compound **1** was carried out using the MMFF94 force field at an energy cutoff of 7.0 kcal/mol and an RSMD threshold of 0.5 Å. A total of thirteen conformations were obtained from the conformational analysis, and seven of which, accounting for more than 1%, were selected for further screening. The seven conformers were relocated and confirmed at the PM6, HF/6-31G(d), and B3LYP/6-1G(d) level to obtain three dominant conformers. Further, the calculation of the theoretical ECD spectra at the B3LYP/6-311G(d, p) level was conducted in MeOH. The final spectrum was obtained by averaging each conformer using the Boltzmann distribution.

### 3.5. Cytotoxic Experiment

CCRF-CEM cells (CL-0058), kindly provided by Procell Life Science & Technology Co., Ltd. (Wuhan, China), were cultured in RIPM-1640 at 37 °C in a humidified atmosphere containing 5% CO_2_ with 10% inactive fetal bovine serum, 2 mM l-glutamine, 100 IU penicillin, and 100 mg/mL streptomycin. The cytotoxicity experiment was conducted using the CCK-8 (Cell Counting Kit-8) assay. Briefly, 4000 cells were seeded on a 96-well plate. After 24 h, different concentrations of the tested compounds were added, and the incubation continued for 48 h. The CCK-8 assay was performed with MD Spectra Max Paradigm.

### 3.6. Ferroptosis Inhibitory Assay

As previously reported [54], 10,000 CCRF-CEM cells were seeded on a 96-well plate for 24 h. Then, ferrostatin-1 (1 μM, as the positive control) and different concentrations of the tested compounds, ranging from 1 μM to 20 μM, were added for first-round screening. After 0.5 h, RSL3 (2 μM) was added to trigger ferroptosis. Four hours later, cellular ATP was detected using the Cell Titer Glo Luminescent Cell Viability assay kit (G7570, Promega, Madison, WI, USA) according to the manufacturer’s instructions. Then, the EC_50_ values were determined as the indicated concentration.

### 3.7. DPPH Assay

According to a reported procedure [55], the stable radical DPPH (2,2-diphenyl-1-picrylhydrazyl) was dissolved in MeOH to a final working concentration of 100 μM. Then, 1 μL of the indicated compounds dissolved in DMSO (10 mM) was added to a final concentration of 100 μM, inverted several times, and allowed to incubate at room temperature for 30 minutes. Samples were then aliquoted to a 96-well microplate and absorbance at 517 nm was recorded using Spectra Max Paradigm (Molecular Devices, San Jose, CA, USA). The relative DPPH normalized to the background (MeOH only) showed the mean ± SD, and the experiments were triplicated.

## 4. Conclusions

Systematic chemical investigation of the deep-sea fungus *Penicillium* sp. 3A00126 yielded 36 compounds, comprising fifteen xanthones (**3**–**15**), six anthraquinones (**16**–**21**), five sesquiterpenoids (**22**–**26**), four diphenyl ethers (**27**–**30**), one polyketone (**31**), four indole alkaloids (**32**–**35**), and one steroid (**36**). Compound **1**, named 11-*O*-acetylaspergillusone B, is a new tetrahydroxanthone, and compound **2**, 7-dehydroxyhuperxanthone A, is a new, fully aromatic xanthone. All 36 isolates were tested for cytotoxicity and ferroptosis inhibitory effects. Sterigmatocystin (**14**) and 5-methoxysterigmatocystin (**15**) showed potent cytotoxicity against CCRF-CEM cells, with IC_50_ values of 5.5 μM and 3.5 μM, respectively, whereas (−)-(7*R*,10*R*)-*iso*-10-hydroxysydowic acid (**26**), verticilatin (**28**), notoamide I (**33**), and notoamide C (**34**) significantly inhibited RSL3-induced ferroptosis, with EC_50_ values of 11.6 μM, 7.2 μM, 11.8 μM, and 2.2 μM, respectively. 

## Figures and Tables

**Figure 1 marinedrugs-21-00234-f001:**
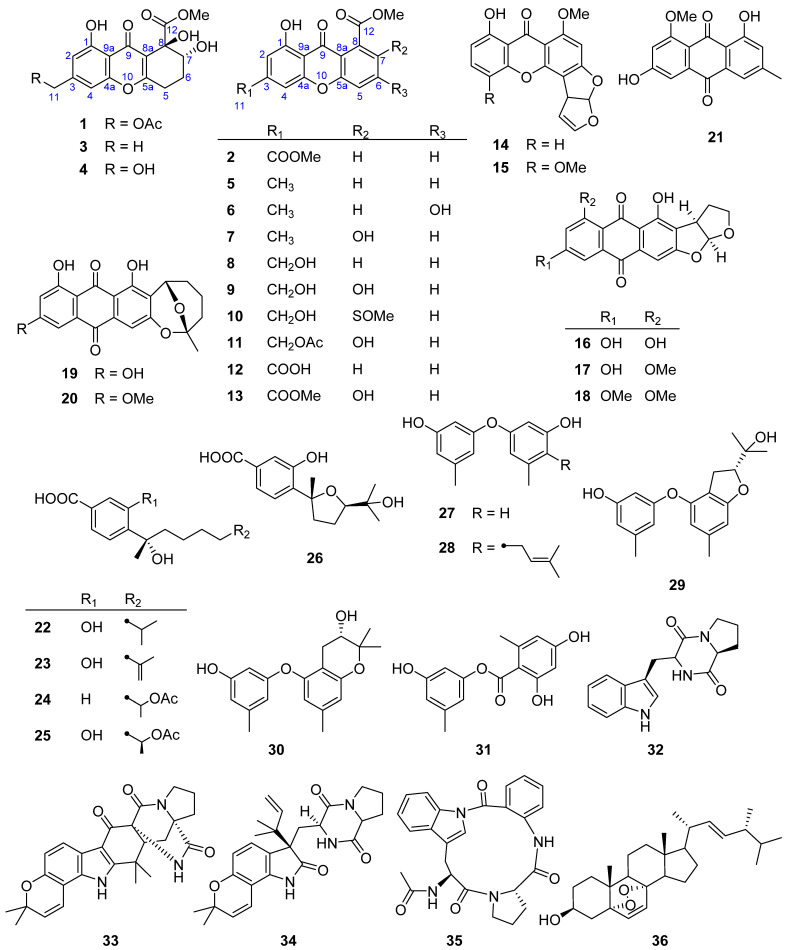
Chemical structures of compounds **1**–**36** isolated from *Penicillium* sp. MCCC 3A00126.

**Figure 2 marinedrugs-21-00234-f002:**
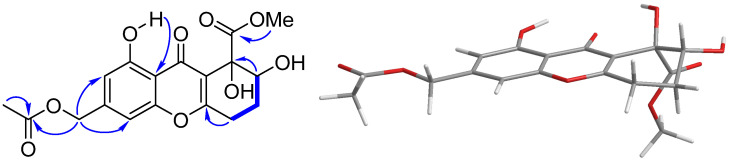
Key ^1^H–^1^H COSY and HMBC correlations and MM2 model of the most stable conformer in the 7,8-anti diol found in **1**.

**Figure 3 marinedrugs-21-00234-f003:**
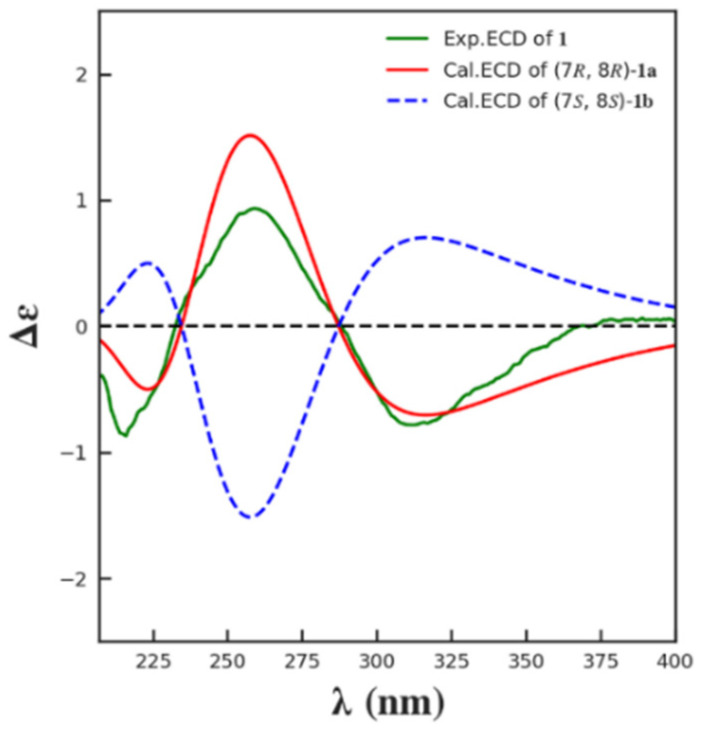
Calculated and experimental ECD spectra of **1**.

**Figure 4 marinedrugs-21-00234-f004:**
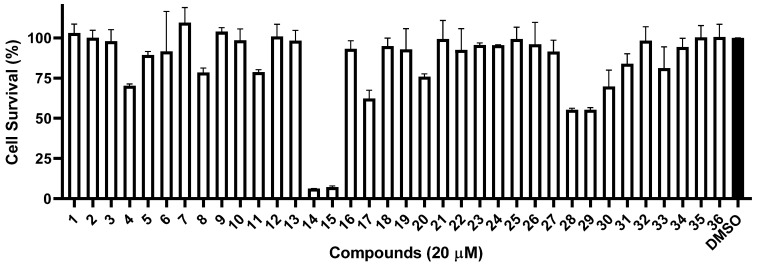
Cytotoxicity of compounds **1**–**36** against CCRF-CEM cells.

**Figure 5 marinedrugs-21-00234-f005:**
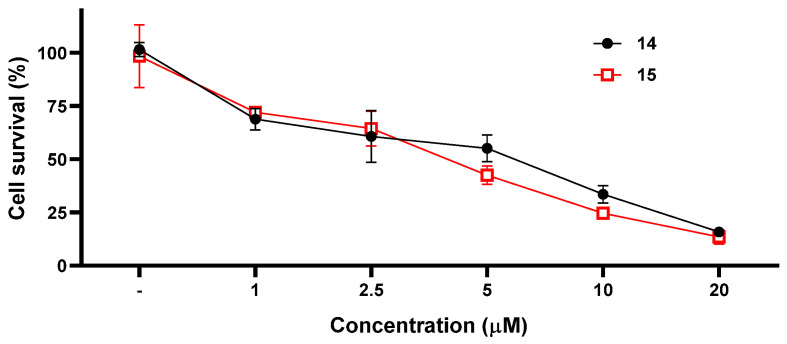
IC_50_ values of compounds **14** and **15** against CCRF-CEM cells.

**Figure 6 marinedrugs-21-00234-f006:**
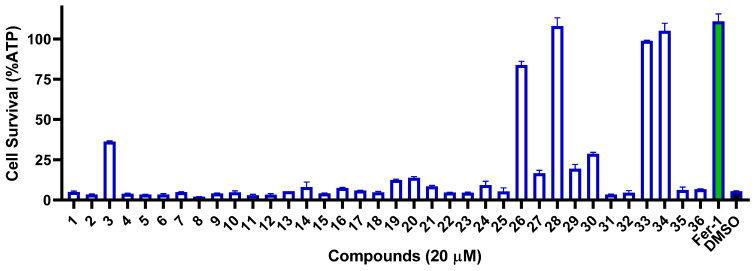
Inhibitory effects of compounds **1**–**36** on RSL3-induced ferroptosis in CCRF-CEM cells.

**Figure 7 marinedrugs-21-00234-f007:**
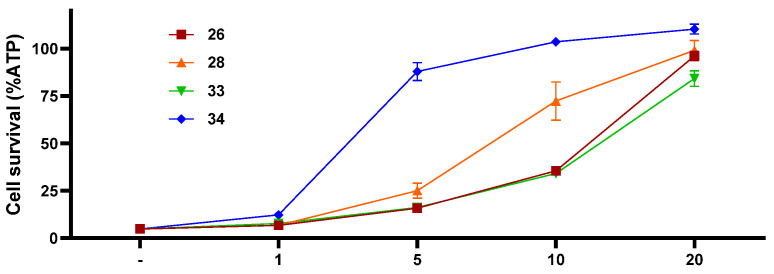
EC_50_ values of compounds **26**, **28**, **33**, and **34** on RSL3-induced ferroptosis in CCRF-CEM cells.

**Figure 8 marinedrugs-21-00234-f008:**
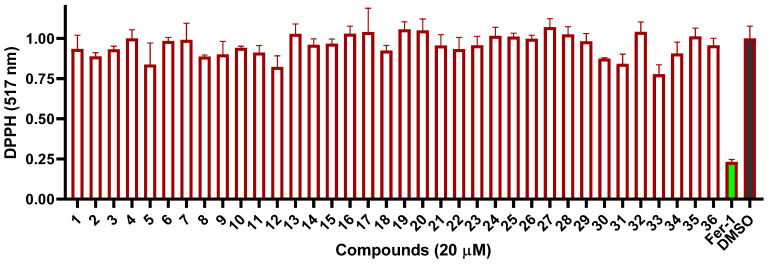
DPPH radical scavenging activities of compounds **1**–**36**.

**Table 1 marinedrugs-21-00234-t001:** ^1^H (400 MHz) and ^13^C (100 MHz) NMR data of **1** and **2** in CDCl_3_ (*δ* in ppm, *J* in Hz within parenthesis).

No.	1		2	
	*δ* _C_	*δ* _H_	*δ* _C_	*δ* _H_
1	160.7 C		161.7 C	
2	109.9 CH	6.75 s	111.6 CH	7.44 s
3	145.0 C		137.6 C	
4	105.5 CH	6.86 s	108.1 CH	7.60 s
4a	156.1 C		155.4 C	
5	26.1 CH_2_	2.87 m	119.6 CH	7.59 (d, 8.0)
6	24.2 CH_2_	2.25 m	135.5 CH	7.81 (dd, 8.0, 7.2)
7	72.6 CH	4.09 (dd, 10.3, 3.6)	123.1 CH	7.36 (d, 7.2)
8	76.2 C		133.7 C	
8a	117.0 C		117.4 C	
9	182.0 C		180.9 C	
9a	109.6 C		111.0 C	
10	167.4 C		156.2 C	
11	65.0 CH_2_	5.13 s	165.3 C	
12	172.7 C		169.2 C	
11-OMe			52.8 CH_3_	3.98 s
11-OAc	170.5 C			
	20.8 CH_3_	2.16 s		
12-OMe	53.4 CH_3_	3.84 s	53.2 CH_3_	4.04 s
1-OH		11.99 s		12.20 s

## Data Availability

Not applicable.

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
