# Peer review of "Ferroptosis Inhibitory Compounds from the Deep-Sea-Derived Fungus Penicillium sp. MCCC 3A00126"

_marinedrugs, 2023, doi:10.3390/md21040234_

Round 1

Reviewer 1 Report

Hao et al. isolated 2 new xanthones and 34 known natural products from the deep-sea -derived fungus Penicillium sp. MCCC 3A00126. Some derivatives showed cytotoxicity while other compounds inhibited RSL3-induced ferroptosis.

Since there are many similar co-isolated compounds, the connectivity and the relative structures seem to be reasonable. I only have some comments and suggestions to the discussion of the absolute configuration and ECD calculation part.

The optical rotation of a similar compound is not a proof for the AC. If the absolute value is small, even for the same compound it should be handled with care. Different concentration or solvent can cause a sign change. Therefore, different substituents can have even larger effect and the sign can be only the basis of a suggestion.

Since the ECD spectra usually contain more than one transition, they provide much more details about the molecule and can be applied safer for the determination of the AC. So it was a good choice to apply ECD. There is however no information provided, how the AC of the related compounds was determined in the literature. It may be important. Is there ECD measurement available also of cpds 3 and 4? If yes, a short discussion would be worth to include.

You provide almost no details about the calculations, except the applied levels. What softwares did you use? How many conformers did you have at the different levels? Were the calculated spectra of the individual conformers similar or different? That can have large impact on the reliability of the computational results. The DFT level chosen is too low for most cases. Here you have analogy and perhaps low flexibility so it may be enough but for more flexible system this level would be surely not enough. Be careful of this in the future.

Concentrations for UV and ECD measurements were not provided.

Although I am not a native English speaker, I felt at some points unnatural usage of the language. A proof reading of a native chemist could be helpful to improve the manuscript.

Reviewer 2 Report

This is another study of the secondary metabolites isolated from a deep-sea-derived fungus of the specie Penicillium. Standard isolation and purification procedures along with a cytotoxic demonstration of some of the metabolites is presented in the paper. 

The structure of compound 1 was elucidated by COSY, HMBC and HSQC experiments. The absolute configuration deduced by ECD theoretical/experimental means. Even though this last step seems enough to demonstrate the configuration of 1, it needs to be completed by two extra information:

a) Computational simulation of the EDC of the (7R,8S) configuration (or their enantiomeric form (7S, 8R)), to compare all the ECD of the 4 possible diasteroisomers at C7and C8.

b) Also trying to run 1D-NOE experiment to relate the position of the OH at C7 and the CO2Me group at C8 could be a great help to confirm the relative location of both functional groups. A worthless 2D-NOESY is presented in the supplementary material (Figure S6) because it was not worked properly. 

c) Although is not needed, DP4+ calculations can discriminate cis or trans configuration at C7 and C8.

Please correct chemical shifts of table 1 according to the data showed in the spectra on the supplementary material. (Eg. 1H chemical shift 6.86, 6.75). Also in the 1H and 13C spectra in this supplementary material show the presence of MeOH as impurity (3.49 ppm and around 50 ppm in 13C and DEPT-135 experiments). 

The structure of compound 2 is briefly explained just in one paragraph. A good description is needed to understand how the HMBC experiment discriminates two possible arrangements of the two functional groups attached at C1 and C3. Biosynthetic arguments can be used in order to discriminate both structures.

Reviewer 3 Report

This review concerns the article type manuscript, entitled Ferroptosis Inhibitory Compounds from the Deep-Sea-Derived Fungus Penicillium sp. MCCC 3A00126, submitted to Marine Drugs journal (Manuscript ID: marinedrugs-2113175).

The article describes isolation, identification, and determination of some biological activity of the compounds obtained from fungus Penicillium taken from the East Pacific Ocean at a depth below 5 thousands meters. This is interesting and corresponds to the aims of Marine Drugs journal.

Two new compounds (denoted as 1 and 2) with the carbon skeletons of xanthene (dibenzo-4H-pyran-4-one) were discovered and their structures were determined mainly using HRESIMS as well as 1D and 2D NMR spectra. All compounds, including those previously determined in literature are presented in a comprehensive manner in Figure 1. The spectral characteristic of the news compounds (1) and (2) is given in Supplementary Materials.

In my opinion the submitted work can be accepted for publication.

Nevertheless, the following remarks should be considered in the revised version.

1.       … from the East Pacific Ocean

2.       Table 1. It should be explained why multiplicity of the 13C NMR signals is given. In other case, it should be removed. Please, add 7-OH and 8-OH signals.

3.       Compound 2. The molecular formula is C17H12O7 (see Figure S13), please correct the text. The DEPT spectrum does not indicate five methylene groups. Please, correct.

4.       Figure S1. The positions of 7-OH and 8-OH groups is not given or not marked on the structure. Please, correct.

5.       Figure S9. The position of the ester carbonyl groups are not given in the structure. Please, correct.

Round 2

Reviewer 1 Report

The authors amended the reviewer's comments and improved the manuscript. Some further Enlish polishment may be needed at the proofreading stage and also please check "B3LYP/6-1G(d)" in the ECD calculation chapter, since there is no 6-1G(d) basis set. It should be maybe 6-31G(d).

Reviewer 2 Report

Authors just add some extra paragraph about the stereochemistry problem around the stereogenic centers C7 and C8.

It is not a straightforward issue to demonstrate the large coupling between H7 and H6 (axial). In fact a simple conformational search made in Maestro software of the two possible syn-7,8-diol and anti-7,8-diol (see attached model) shows 5 conformers each in 20 kJ/mol range.

In the case of the sin-7,8-diol two conformers in 0.8 kJ/mol showed small couplings and another 3 conformers showed large coupling. Taking into account the Boltzmann weighted contribution of the five conformers, two small coupling constants(JH7-H6a and JH7-H6e) are experimentally found.

On the other hand for the anti-7,8-diol 2 major conformers in 2.7 kJ/mol accounting for the 95% of the conformer-population showing an axial-axial large coupling similar to the reported by the authors.

Yes, indeed! the coupling constants can explain the configuration, but this discussion should be presented in the structural elucidation of 1. 

Reviewer 3 Report

The revised version takes into account all my previous comments. Therefore, the work may be considered for publication.

Round 3

Reviewer 2 Report

A new paragraph was added in the discussion of the relative configuration of C7 and C8:

The coupling constants between H-7 and H2-6 of 1 (J = 10.3 Hz, 3.6 Hz) indicated the same pseudoaxial position of H-7 as that of 4 [7], which was confirmed by the NOESY correlation of H-7 to H-5 (Figure 2).....

The NOESY correlation between H7 and H5 (which are reported both axial and equatorial protons at the same chemical shift, 2.87 ppm) does not confirm the relationship between H7 and H5!

Axial or equatorial protons? This NOE cross peaks does not add information to the relative configuration. Please leave the discussion as follows: 

The coupling constants between H-7 and H2-6 of (= 10.3 Hz, 3.6 Hz) indicated the same pseudoaxial position of H-7 as that of  [7, as it was found in a simple MM2 conformational study of both possible 7,8-anti and 7,8-syn diols

Please, remove the NOESY arrow on Figure 2 and change the legend of this figure to: Key 1H-1H COSY and HMBC correlations and MM2 more stable conformer in the 7,8-anti diol found in 1. 

In the supplementary material, NOESY on figure S6 should phase and baseline corrected!

Round 4

Reviewer 2 Report

All my suggestions have been addressed, therefore under my criteria the paper now meets the requirements to be published.